



# Arctic mixed-phase clouds sometimes dissipate due to insufficient aerosol: evidence from observations and idealized simulations

Lucas Sterzinger[1], Joseph Sedlar[2,3], Heather Guy[4,5], Ryan R. Neely III[4,5], and Adele L. Igel[1]

[1]University of California, Davis, Davis, California
[2]Cooperative Institute for Research in Environmental Sciences, University of Colorado Boulder, Boulder, Colorado
[3]NOAA/Global Monitoring Laboratory, Boulder, Colorado
[4]National Centre for Atmospheric Science, Leeds, U.K.
[5]School of Earth and Environment, University of Leeds, Leeds, U.K.

**Correspondence:** Lucas Sterzinger (lsterzinger@ucdavis.edu)

**Abstract.** Mixed-phase clouds are ubiquitous in the Arctic. These clouds can persist for days and dissipate in a matter of hours. It is sometimes unknown what causes this sudden dissipation, but aerosol-cloud interactions may be involved. Arctic aerosol concentrations can be low enough to affect cloud formation and structure, and it has been hypothesized that, in some instances, concentrations can drop below some critical value needed to maintain a cloud.

We use observations from a Department of Energy ARM site on the north slope of Alaska at Oliktok Point (OLI), the ASCOS field campaign in the high Arctic Ocean, and the ICECAPS-ACE project at the NSF Summit Station in Greenland (SMT) to identify one case per site where Arctic boundary-layer clouds dissipated coincidentally with a decrease in surface aerosol concentrations. These cases are used to initialize idealized large eddy simulations (LES) in which aerosol concentrations are held constant until, at a specified time, all aerosols are removed instantaneously – effectively creating an extreme case of

aerosol-limited dissipation which represents the fastest a cloud could possibly dissipate via this process. These LES simulations are compared against the observed data to determine whether cases could, potentially, be dissipating due to insufficient aerosol. The OLI case's observed liquid water path (LWP) dissipated faster than its simulation, indicating that other processes are likely the primary driver of the dissipation. The ASCOS and SMT observed LWP dissipated at similar rates to their respective simulations, suggesting that aerosol-limited dissipation may be occurring in these instances.

We also find that the microphysical response to this extreme aerosol forcing depends greatly on the specific case being simulated. Cases with drizzling liquid layers are simulated to dissipate by accelerating precipitation when aerosol is removed while the case with a non-drizzling liquid layer dissipates quickly, possibly glaciating via the Wegener-Bergeron-Findeisen (WBF) process. The non-drizzling case is also more sensitive to INP concentrations than the drizzling cases. Overall, the simulations suggest that aerosol-limited cloud dissipation in the Arctic is plausible and that there are at least two microphysical

pathways by which aerosol-limited dissipation can occur.



## 1 Introduction

The Arctic has been shown to be extremely sensitive to a warming climate, with data showing the Arctic warming anywhere from 1.5 - 4.5x the global mean warming rate (Holland and Bitz, 2003; Serreze and Barry, 2011; Cohen et al., 2014; Previdi et al., 2021). Clouds, in general, directly affect the surface energy budget and can act as net-warming or net-cooling influences, depending on their specific physical characteristics. Of particular note in the Arctic environment are low-level, boundary layer stratocumulus clouds which cover large fractions of the Arctic throughout the year (Shupe, 2011). They have been found to be a net-warming influence on the surface, except for a short period in the summer when they act as a net-cooling influence (Intrieri et al., 2002; Shupe and Intrieri, 2004; Sedlar et al., 2011). These clouds tend to be mixed-phase, meaning they simultaneously contain liquid and ice water. Shupe et al. (2006) found that mixed-phase clouds accounted for 59% of the clouds identified during a year-long campaign on an icepack in the Beaufort Sea, with the remaining 41% consisting of mostly ice-only clouds. Difficulties in parameterizing ice processes, the physical complexities and uncertainties involved with liquid and ice water coexisting, and a lack of observations in the Arctic make these clouds a known problem for numerical models of all scales (Sotiropoulou et al., 2016; Klein et al., 2009; Morrison et al., 2009, 2012, 2011); understanding the processes involved in the formation and dissipation of these clouds is essential to understanding the energy balance in the Arctic and for proper representation in models.

These Arctic mixed-phase boundary layer clouds often last for days at a time, and dissipate in a matter of hours (Shupe, 2011; Morrison et al., 2012). This persistence is surprising given the inherit microphysical instability of mixed-phase clouds, which can be affected by the Wegener-Bergeron-Findeisen (WBF) process (Wegener, 1911; Bergeron, 1935; Findeisen, 1938) in which, if the environmental vapor pressure is between the saturation vapor pressure of liquid and ice water, ice grows via deposition at the expense of liquid. Without processes maintaining high supersaturations with respect to water, the WBF process could glaciate (i.e. completely convert to ice) the cloud. The mechanisms behind Arctic mixed-phase clouds' persistence and rapid dissipation are not well known (Morrison et al., 2012). Mauritsen et al. (2011) hypothesized that the low cloud condensation nuclei (CCN, a subset of the available aerosol with radii generally between ∼1 nm - 0.5 $\mu$m, required for cloud droplet formation) concentrations in the Arctic could have an effect on cloud dissipation, and coined the term "tenuous cloud regime" to describe clouds whose structures are limited by aerosol availability, and showed that aerosol concentrations over the central Arctic ice pack are often observed to be low enough to affect cloud formation.

Modeling studies (Birch et al., 2012; Stevens et al., 2018; Sotiropoulou et al., 2019) have supported the existence of the tenuous cloud regime, but none of these studies has focused directly on the role of limited aerosol on the dissipation of Arctic mixed-phase boundary layer clouds.

Morrison et al. (2012) presented an overview of the long-term persistence of mixed-phase Arctic clouds. These clouds are maintained by cloud-scale updrafts which, if strong enough, create conditions where the environment is supersaturated with respect to both liquid water and ice; in this situation, both liquid droplets and ice crystals will grow, and the WBF process is not active (Korolev, 2007). High concentrations of supercooled liquid water droplets at cloud-top will cool radiatively, creating a buoyant overturning circulation that can further enhance the cloud (Brooks et al., 2017). Moisture inversions are also very



common at the top of the Arctic boundary layer, occurring upwards of 90% of the time in the winter months and 70-80% in the summer (Naakka et al., 2018; Egerer et al., 2020; Sedlar et al., 2012; Devasthale et al., 2011). The presence of a moisture inversion near cloud-top can act as a source of water vapor through cloud-top entrainment (Solomon et al., 2011; Sedlar et al., 2012; Sedlar and Tjernström, 2009). The combination of cloud-top moisture entrainment and a cloud-scale buoyant overturning circulation allow Arctic mixed-phase clouds to persist even with low surface heat and moisture fluxes; if the boundary layer is

decoupled from the surface, the moisture inversion may be the only source of moisture for a cloud (Sedlar et al., 2012; Brooks et al., 2017).

Unlike low-level clouds at lower latitudes, Arctic boundary layer clouds generally warm the surface (Shupe and Intrieri, 2004). Low-level Arctic clouds have a similar albedo to the ice surface, which means that the shortwave cooling effect (whereby clouds act to cool the surface by reflecting a higher proportion of the incoming solar radiation) is negligible. In the summer

months where more of the Arctic surface has melted from ice to open water or melt ponds, the surface albedo is lower and the shortwave cooling effect of the cloud dominates (Intrieri et al., 2002; Tjernström et al., 2014).

The availability of atmospheric aerosol, some of which serve as CCN, has direct affects on cloud properties. An increase in CCN concentration, while keeping the amount of precipitable water constant, increases cloud albedo by dividing the available water vapor between more activated CCN, resulting in more, but smaller, cloud droplets. This shift from fewer large droplets

to more numerous small droplets results in a cloud that is more reflective to shortwave radiation, a phenomenon known as the Twomey effect (Twomey, 1977). The resulting reflectance of shortwave radiation causes increased surface cooling, but this effect competes with increased emissivity of the cloud also caused by the increase in CCN (for thin clouds not already emitting as a blackbody; less than $\sim 40$ g m$^{-2}$ LWP) (Garrett et al., 2002; Garrett and Zhao, 2006; Loewe et al., 2017).

While the effects of aerosols and CCN on cloud properties are a focus of much scientific investigation, outside of some

recent studies (Mauritsen et al., 2011; Loewe et al., 2017; Stevens et al., 2018; Guy et al., 2021) little has been done to examine the effect of abnormally low aerosol concentrations on clouds. Mauritsen et al. (2011) proposed, through observation, the existence of a tenuous cloud regime where cloud structure is limited by CCN concentration (which, in lower latitudes, often ranges from 100 - 1000 cm$^{-3}$, but has been observed as low as 1 cm$^{-3}$ in the Arctic, e.g. Jung et al. 2018). Chandrakar et al. (2017) performed a lab study with a cloud chamber in which aerosol were removed from a turbulent, cloudy environment.

They found that after aerosol injection was turned off, the cloud did not appreciably change until near the 1-hour mark. At this point, interstitial aerosol were sufficiently removed and cloud dissipation occurred rapidly within the following 30-40 minutes.

Sedlar et al. (2021) found that surface aerosol concentrations at Utqiaġvik, Alaska were similar before and after cloud dissipation in the winter and spring months, and slightly higher after dissipation in the summer and fall months, when looking at statistics of instrument retrievals from 2014-2018, suggesting that limited aerosol is not a primary method of cloud dissipation.

However, this study approached this question climatologically, and we believe that if aerosol-limited dissipation does occur, it is infrequent enough to be hidden in these types of analyses. Furthermore, Sedlar et al. (2021) focused solely on measurements from a single site (Utqiaġvik) which may be more polluted than the rest of the Arctic due to increased human activity. At more remote locations, aerosol-limited dissipation may occur more frequently.




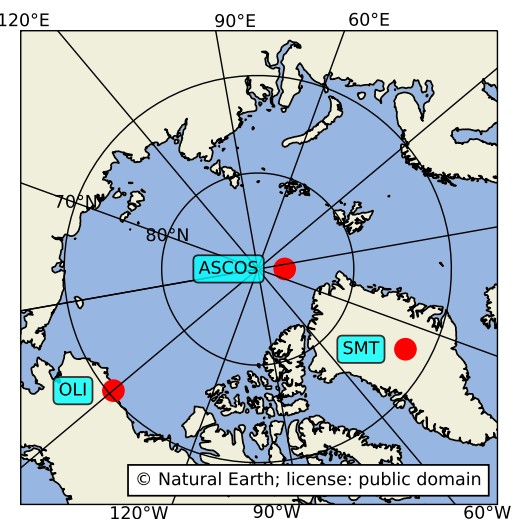

**Figure 1.** Map showing the locations of data taken from OLI, ASCOS, and SMT

In this study, we investigate whether or not aerosol-limited dissipation occurs on a case-by-case basis. While likely infre-
quent, this method of cloud dissipation is worth examining in more detail because of how sensitive the Arctic environment is
to low-level cloud cover, and the highly uncertain changes in Arctic aerosol concentration (both natural and anthropogenic) in
a warming climate (e.g. Schmale et al., 2021). We examine three observed cases of potential aerosol-limited dissipation across
three different environments (northern Alaskan coast, high Arctic pack ice, and the Greenland ice sheet) and use large eddy
simulations (LES) to simulate a "worst-case scenario" of aerosol-limited dissipation: immediately removing all aerosols from
a simulated cloudy environment and comparing changes in cloud properties to observations, which should indicate whether or
not these cases should continue to be investigated as examples of this phenomenon.

## 2 Methods

### 2.1 Case Overviews

Observations from the DOE ARM Site at Oliktok Point, Alaska (OLI; $71.32°$ N $156.61°$ W, 2 m ASL), the 2008 Arctic
Summer Cloud Ocean Study campaign (ASCOS; $87.19°$ N $9.67°$ W, 0 m ASL), and the ICECAPS-ACE project at Summit
Station in Greenland (SMT; $72.6°$ N, $38.5°$ W, 3250 m ASL) were used to identify cases where cloud dissipation was observed
coincidentally with a decrease in surface aerosol concentration. For simplicity, we focus solely on single-layer, low-level,

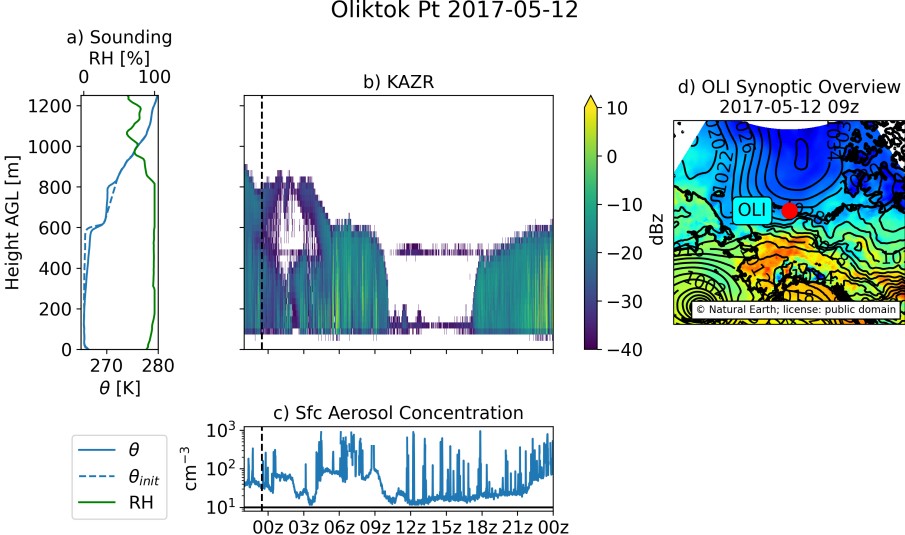

**Figure 2.** Overview of observations for Oliktok Point, AK (OLI): **a)** potential temperature ($\theta$, blue) and relative humidity (green) from a sounding at 23:30 UTC on 2017-05-11, **b)** KAZR (radar) general mode reflectivity (dBz), **c)** surface aerosol concentration (CPC-fine), **d)** ERA5 reanalysis mean sea level pressure (contours) and 2 meter temperature (color shading) for 2017-05-12 at 09:00 UTC. Dashed vertical line on panels (b-c) represent the time at which the sounding was taken. Solid horizontal line on panel (c) represents a surface aerosol concentration of $10 \ \mathrm{cm}^{-1}$. For OLI alone, $\theta_{init}$ in panel (a) represents the profile used for model initialization.

boundary layer mixed-phase clouds. Figure 1 shows the three case locations on a map. Details of each case are summarized in Figures 2-4 and discussed briefly in the sections below.

### 2.1.1 Oliktok Point

From 2013-2021, the United States Department of Energy's Atmospheric Radiation Measurement (ARM) user facility operated a mobile facility at Oliktok Point, Alaska (henceforth OLI), located on the northern Alaskan coastline 260 km southeast of the permanent ARM facility in Utqiaġvik. We analysed data from Oliktok Point between 2016-2019 to find periods in which surface aerosol concentrations via a condensation particle counter (CPC) (measuring particles 10-3000 $\mu$m; Kuang et al., 2016) were observed to decrease from $> 50 \ \mathrm{cm}^{-3}$ to $< 20 \ \mathrm{cm}^{-3}$ in a span of 4 hours. Many such periods exist, and the results were examined manually to select cases where aerosol-limited dissipation may have been a factor in transitioning from a cloudy to cloud-free environment.

One such case (Fig. 2) occurred on the 12th of May, 2017. At 09:00 UTC, the CPC measured a transition in aerosol concentration from $\sim 100 \ \mathrm{cm}^{-3}$ to $<10 \ \mathrm{cm}^{-3}$ in the span of about one hour (Fig. 2c). Aerosol data from OLI was particularly noisy, with a clear trend of concentrations $\sim 100 \ \mathrm{cm}^{-3}$ but with intermittent spikes upwards of $1000 - 10000 \ \mathrm{cm}^{-3}$ (not shown). To smooth out the data and best show what we consider to be a representative aerosol concentration timeseries, we filtered out values $> 1000 \ \mathrm{cm}^{-3}$ and downsampled the result from one-second to one-minute averages. Data from the Ka-band ARM

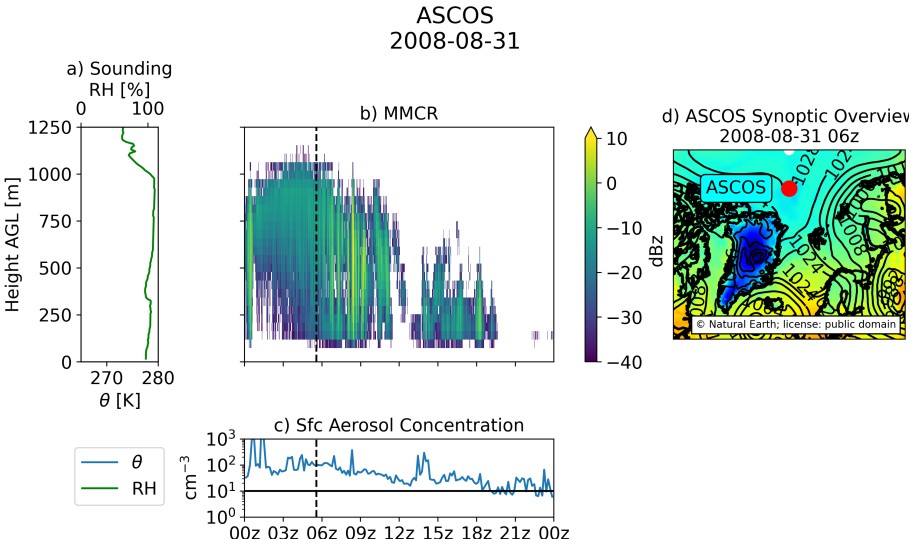

**Figure 3.** Same as Figure 2 for the ASCOS case.

zenith radar (KAZR; Lindenmaier et al., 2015) show a cloud with a top around $700 - 750$ m transition to clear skies at the same time (Fig. 2b). Around 18:00 UTC, aerosol concentrations begin to increase and a new cloud is visible on radar.

The sounding from OLI (Fig. 2a) shows a well-mixed, surface-coupled boundary layer with a capping inversion at 600 m. A second, smaller inversion can be seen at 800 m. However, since this balloon launch occurred approximately nine hours before cloud dissipation and the radar (Fig. 2b) indicates that the second, higher, cloud layer descends and merges with the lower layer, this second inversion is removed prior to model initialization ($\theta_{init}$; Fig. 2a, dashed line). Relative humidity (RH) is high ($> 90\%$) throughout the boundary layer, as well as up to 200 m above the cloud layer before it starts decreasing above 800

m. Aerosol concentrations fluctuate between the time of the sounding and the time of dissipation, so a value of 80 cm$^{-3}$, as a representative average of the pre-dissipation concentration, is used to initialize the simulation.

    Surface analysis from ERA5 reanalysis (European Centre for Medium-Range Weather Forecasts, 2019) at 09:00 UTC on May 12, 2017 (Fig. 2d) depicts a high pressure system in the Beaufort Sea north of Oliktok Point and a stationary front situated along the Brooks Range of mountains more than 200 km inland. A weak ridge of high pressure extends from the high pressure

system to Oliktok Point, with weak low pressure areas identified to the southeast and southwest of Oliktok Point. This pressure ridge is not present in analysis maps $\pm 6$ h from 09:00 UTC. Analysis of pressure, temperature, and wind timeseries (not shown) at OLI suggests a weak frontal passage with a temperature drop and wind shift near the same time as cloud dissipation, and a constantly decreasing surface pressure throughout the entire 24 h period.

### 2.1.2   ASCOS

The ASCOS field campaign (Tjernström et al., 2014) took place mostly during the month of August 2008 with a focus on observing and understanding Arctic low-level clouds and improving their representation in climate models. From 12 August





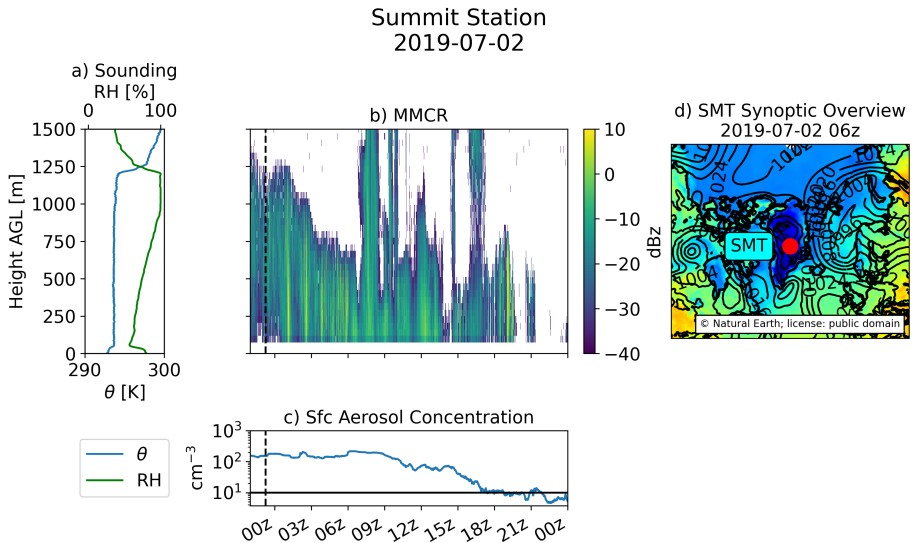

**Figure 4.** Same as Figures 2-3 for the SMT case.

through 1 September, the Swedish icebreaker *Oden* was purposefully trapped in (and drifted with) an ice floe in the high Arctic ocean (87° N) north of the island of Svalbard (Fig. 1).

On the 31st of August, 2008, *Oden* was trapped in an ice floe at 87.19° N 9.67° W. A sounding from this time (Fig. 3a) shows

a slightly stable layer up to 400 m and a well-mixed layer from 400 m to cloud-top at 1,000 m. Millimeter-wave cloud radar (Clothiaux et al., 2000) reflectivities show a low-level cloud layer (Fig. 3b), which had persisted for approximately 1 week prior, dissipated coincidentally with a decrease in surface aerosol concentration from >100 cm$^{-3}$ to <10 cm$^{-3}$ (Sedlar et al., 2011; Mauritsen et al., 2011; Sotiropoulou et al., 2014). aerosol concentrations were collected using a differential mobility particle sizer (DMPS; Birmili et al. 1999; Tjernström et al. 2014) measuring size distributions of particles between 3 nm -

10 $\mu$m. Helicopter flights measuring aerosols during this time found that concentrations were below 10 cm$^{-3}$ (for aerosols < 14 nm) for the entirety of the boundary layer (Stevens et al., 2018) during a flight in the dissipation period at 20:13 UTC. After cloud dissipation, winds which were previously calm were observed to more consistently blow from the northeast (not shown). At the same time, surface temperature drops over 6 °C, though it's unclear whether there was a change in airmass or if temperature dropped as cloud is no longer present as a warming influence on the surface. Surface pressure analysis (Fig. 3d)

shows the extension of northern high pressure directly over the location of *Oden* at this time, suggesting a possible change in airmass.

Like the OLI case, RH values are generally high throughout the boundary layer. Unlike OLI, there is a dry layer at 400 m. A change in $\theta$ and RH profiles at 400 m indicate weak decoupling at this level. This case has previously been investigated as existing in a potentially tenuous regime (Mauritsen et al., 2011; Loewe et al., 2017; Stevens et al., 2018; Tong, 2019).





### 2.1.3 Greenland


A third case was observed on July 2nd, 2019 at the National Science Foundation Summit Station (henceforth SMT) during the ICECAPS-ACE project. ICECAPS-ACE (Integrated Characterization of Energy, Clouds, Atmospheric State, and Precipitation at Summit - Aerosol Cloud Experiement; Shupe et al., 2013; Guy et al., 2021) consists of a suite of instruments for measuring atmospheric properties (including surface aerosol concentrations) at SMT (3250 m ASL, 72.6N, 38.5 W).

Figure 4 shows an overview of this case, which was first reported in Guy et al. (2021) as an potential example of aerosol-limited dissipation. A well-mixed boundary layer topped with a single cloud layer approximately 200 m in thickness $\sim$1200 m above the surface ($\sim$4400 m ASL) is observed with a balloon sounding the previous day (2019-07-01 at 23:16 UTC, Fig. 4a). Radar data (Fig. 4b) from a millimeter wave cloud radar (Bharadwaj, 2010) show the cloud top lowering before a transition to clear-sky as surface aerosol concentrations decrease (CPC measured particles >5 nm in diameter in 1-minute intervals; Guy
et al. 2020) from 200 cm$^{-3}$ to < 10 cm$^{-3}$ (Fig. 4c) in a period of 9 hours. Synoptic conditions at this time show that this case occurred during a period of anomalously low 500 hPa heights and above average winds from the southeast (Guy et al., 2021, not shown). A small inversion near the surface decouples the boundary layer from surface fluxes of heat and moisture. Unlike OLI and ASCOS, below-cloud RH decreases towards the surface, reaching $\sim$50% directly above the surface inversion.

### 2.2 Model Description

The Colorado State University Regional Atmospheric Modeling System (RAMS; Cotton et al., 2003) was used to run large-eddy resolving simulations (LES) of each case. The RAMS model has been shown to perform well at LES scales (e.g. Cotton et al., 1992; Jiang et al., 2001; Jiang and Feingold, 2006). Radiation parameterization is provided by the Harrington scheme (Harrington, 1997), and turbulence is parameterized by a Deardorff level 2.5 scheme, which parameterizes eddy viscosity as a function of turbulent kinetic energy (TKE).

RAMS uses a double-moment bulk microphysics scheme (Walko et al., 1995; Meyers et al., 1997; Saleeby and Cotton, 2004) that predicts the mass and number concentration of eight hydrometeor categories: cloud droplets, drizzle, rain, pristine ice, aggregates, snow, hail, and graupel. Each of these hydrometeor categories is represented by a generalized gamma distribution. The scheme simulates nucleation (cloud and ice), vapor deposition, evaporation, collision-coalescence, melting, freezing, secondary ice production, and sedimentation. Cloud droplets are activated from aerosol particles using lookup tables (Saleeby
and Cotton, 2004) built based on Köhler theory and cloud droplet growth equations formulated in Pruppacher and Klett (1997). Water vapor is depleted from the atmosphere upon activation by assuming that newly activated droplets have a diameter of 2 $\mu$m.

Ice crystals are heterogeneously nucleated by the parameterization in DeMott et al. (2010), with the number of ice nuclei ($L^{-1}$) given by:

$$n_{in} = a(273.16 - T_k)^b (n)^{c(273.15 - T_k) + d} \tag{1}$$

Where $n_{in}$ is the ice nuclei number concentration, $T_k$ is the air temperature in Kelvin, and $a, b, c, d$ are constants. The variable $n$ in the original DeMott parameterization is the number concentration of aerosol particles with diameters larger than 0.5 $\mu$m,





**Table 1.** Case names and abbreviations along with the initial aerosol concentration ($n_{aer}$), ice nuclei concentration (IN), and aerosol removal time.

| Name | Abbr. | $n_{aer}$ | IN ($n$ in eq. 1) | Aerosol Removal Time |
|---|---|---|---|---|
| Oliktok Point | OLI | 80 cm$^{-3}$ | 5 L$^{-1}$ | 09:00 UTC |
| ASCOS | ASCOS | 89 cm$^{-3}$ | 5 L$^{-1}$ | 06:00 UTC |
| Greenland Summit Station | SMT | 200 cm$^{-3}$ | 0.1 L$^{-1}$ | 06:00 UTC |

but in this study we have elected to use an option in RAMS to set a constant $n$ at model runtime (values of $n$ used are noted in Table 1).

## 2.3 Experiment Setup

The observations were used to generate an initial sounding and to specify aerosol concentration for each simulation. For each case, RAMS was run with a horizontal domain of 6x6 km$^2$ with a spacing of 62.5 meters and vertical spacing of 6.25 meters with a domain height of 1250 m (200 levels) for OLI and ASCOS, and 1600 m domain height (256 levels) for SMT (to accommodate the deeper boundary layer). Lateral cyclic boundary conditions are employed, allowing features that pass through one side of the domain to emerge from the other. While soundings are available at all measurement sites, they do not contain information on the liquid water/ice content of the cloud. To properly initialize RAMS, liquid water was manually added to sounding data. In the absence of observed vertical profiles of liquid water content, a linear profile of water mass (zero at cloud base and maximum at cloud-top) was added, with a slope chosen such that integrating the liquid profile from cloud base to cloud-top yielded the observed liquid water path.

We use a simplified aerosol treatment in which number concentrations are fixed to a single value throughout the domain. Aerosol are not depleted, rather the prescribed number concentration acts as an upper bound on the number of activated cloud droplets allowed in a given grid cell. For each of the cases, we performed simulations in which all aerosol are instantaneously and permanently removed from the environment. We do this in order to simulate the fastest possible cloud dissipation; if the observations show faster dissipation than we simulate, then we can conclude that the observed dissipation was not driven entirely by a lack of aerosol particles. Prior to this aerosol removal time, a temperature nudging scheme is used to maintain a stable cloud. At each time step, each grid point is linearly nudged back to the initial temperature profile with a time scale $\tau = 1$ h. Nudging values are computed based on the current domain-average temperature profile, so all grid points at a given height $z$ are nudged the same amount. The result in all simulations is a cloud that is quasi-steady in thickness and water content. After the removal of aerosol from the model, the temperature nudging scheme is turned off and the thermodynamics of the system are allowed to evolve naturally - this is done so that the post-aerosol environment is able to evolve naturally. Large-scale subsidence is applied throughout the simulation by imposing a horizontal divergence of $2 \times 10^{-6}$ s$^{-1}$ at every model level, with a boundary condition of $w_{sub} = 0$ at the surface.





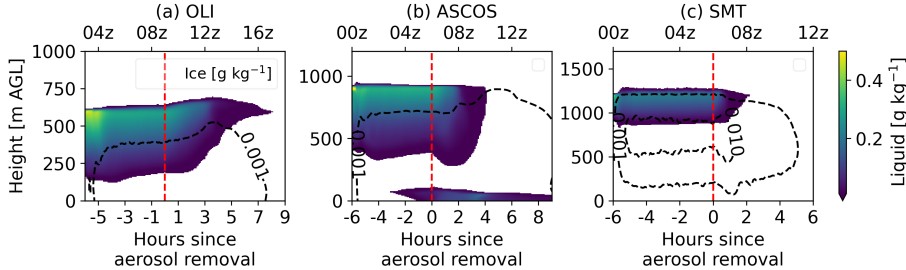

**Figure 5.** Contours for cloud water (color shading) and ice (dashed) **(a)** OLI, **(b)** ASCOS, and **(c)** SMT simulations. Ice is contoured at 0.01 and 0.001 g kg$^{-1}$ [the 0.01 g kg$^{-1}$ contour is only present in **(c)**].

A list of experiments and initial aerosol/ice nuclei concentrations is found in Table 1. Measurements of surface aerosol concentrations were used to initialize the aerosol concentration for each simulation. For ice nuclei (IN) concentrations, we

performed sensitivity tests to different values of the $n$ in the IN parameterization (equation 1) and found that in OLI and ASCOS there was little change in the liquid water for $n$ = 1, 5, or 10 L$^{-1}$. There were moderate differences in ice water content, and as there are ice water path (IWP) retrievals for both the OLI and ASCOS cases, we picked a value of $n$ that yielded simulated IWP values closest to observations. For the SMT case, no ice measurements were available. Both simulated ice and liquid were sensitive to choice of $n$, so a value of 0.1 L$^{-1}$ was used; this value is consistent with currently unpublished INP data from

Summit Station (available upon request) and resulted in simulated liquid water path that was closest to observations.

## 3 Results

Figure 5 shows domain-averaged liquid water (color shading) and ice (dashed contours at 0.01 and 0.001 g kg$^{-1}$) show typical Arctic mixed-phase clouds in which a layer of supercooled liquid water is situated at cloud top with ice precipitating below. In OLI and ASCOS, the liquid layer is well-above the ice layer ($\sim$200 m from cloud top to the 0.001 g kg$^{-1}$ ice contour),

whereas in SMT the ice extends nearly to cloud top.

Figure 6 shows the domain-mean liquid water path (LWP) for the OLI, ASCOS, and SMT simulations and the corresponding observed LWP. Observed LWP data were taken from microwave radiometers at OLI (Gaustad, 2014), ASCOS (Westwater et al., 2001), and SMT (Cadeddu, 2010) This figure shows that, in all cases, the simulated LWP decreases to near-zero within hours of the aerosol removal time (09z in OLI, 06z in ASCOS and SMT). Both the OLI and ASCOS simulations show a slow LWP

response to aerosol removal, with LWP approaching 0 g kg$^{-1}$ in about 4-5 hours. The SMT simulation, on the other hand, has a very pronounced LWP response to aerosol removal, with LWP approaching zero within 2 hours. With instantaneous aerosol removal, the simulations should theoretically represent the fastest possible dissipation of a cloud due to insufficient aerosol. Where this simulated LWP response is slower than observations - such as OLI - it is likely that a lack of aerosol is not in fact the primary driver of dissipation. Where the simulated LWP response is more similar to observations (ASCOS and SMT), it is

more likely that these are indeed cases of aerosol-limited dissipation.



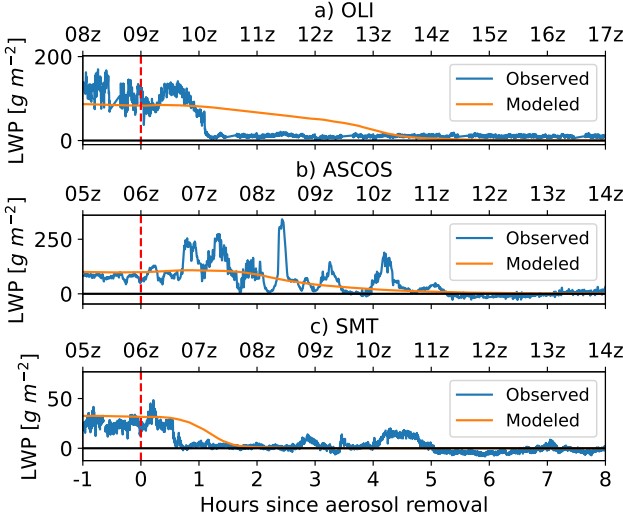

**Figure 6.** Liquid water path evolution for observations (blue) and modeled domain-averages (orange) at **(a)** OLI, **(b)** ASCOS, and **(c)** SMT. The red dashed line denotes the time at which aerosol were removed from the simulations.

Each case will now be discussed in detail; since the time of aerosol removal was determined rather subjectively, and because the aim of this paper is not to compare directly with observations (but instead to compare timescales), all further discussion will be discussed in the context of hours before/after aerosol removal, instead of UTC, to better compare cases with one another.

## 3.1 OLI

It is evident from Figure 6a that the OLI cloud dissipation was not due to a lack of available aerosol. While the observed LWP decreased from 100 g kg$^{-1}$ to <10 g kg$^{-1}$ in ~1 hour, modeled LWP took 4-5x this time. While the OLI case may not be a real-world example of aerosol-limited dissipation, examining its simulated response to aerosol removal when compared to the different cases still yields valuable insights to this phenomenon.

Domain-average 2D and column-integrated liquid and ice budgets, radiative heating, and vertical momentum flux for OLI
are shown in Figure 7. After a 1-hour spin-up period (not shown), the cloud settles to quasi-equilibrium with approximately constant liquid precipitation reaching the surface and consistently positive integrated cloud droplet growth by condensation, which occurs primarily at cloud base, where supersaturation is largest, and at cloud top. In the cloud interior there is slight net liquid evaporation and net ice depositional growth due to an active WBF process. The growth of ice and liquid are balanced by persistent precipitation of both liquid and ice hydrometeors throughout the pre-aerosol removal time period. Riming makes up
only a small part of the liquid and ice budgets. Radiative cooling (Fig. 7c) is strongest at cloud top as expected, which drives the overturning circulation responsible for maintaining the cloud. Vertical momentum flux ($\overline{w'w'}$; Fig. 7f) is strongest throughout the mixed portion of the boundary layer - from cloud top down to 100 m.



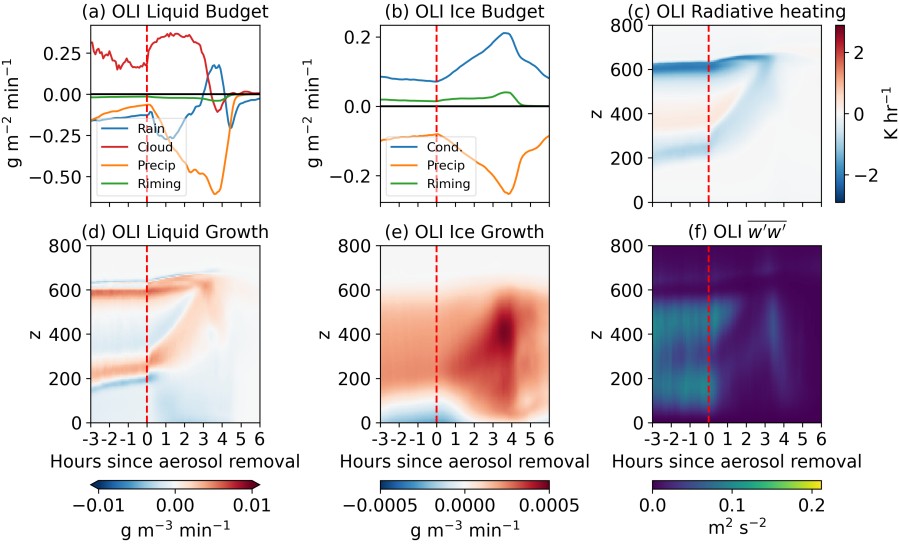

**Figure 7.** Vertically-integrated domain-average budgets for **(a)** liquid and **(b)** ice hydrometeors as well as domain-average **(c)** radiative heating/cooling, 2D growth of **(d)** liquid and **(e)** ice, and **(f)** vertical momentum flux for OLI. Red line denotes time of aerosol removal. Liquid budget **(a)** shows condensational growth of cloud and rain, and removal by precipitation (precip) and riming. Ice budget **(b)** shows growth of all ice species by condensation (cond), riming, and removal from precipitation (precip).

After the removal of aerosol, a large increase in liquid precipitation and a smaller relative increase in ice precipitation occur. Removing aerosol inhibits the nucleation of new cloud droplets, meaning that any supersaturation must be condensed onto existing droplets rather than being used to create new droplets. This results in a rapid increase in droplet sizes (not shown) and an enhanced collision-coalescence process, leading to increased liquid precipitation. The precipitation is initially strongest near cloud base and contributes to a rise in cloud base. Since new droplets are unable to be nucleated (and available liquid to condense upon is being precipitated), supersaturation levels increase (not shown). Approximately three hours after aerosol removal, cloud condensation falls off sharply. Figure 7(d-e) show that, after aerosol removal, there is an increase in ice growth which maximizes after liquid is mostly removed (Fig. 6a). However, at this point the cloud top radiative cooling has ceased, circulations weaken, and the ice begins to slowly decay as well.

Figure 5a shows that, after aerosol removal, the OLI simulation dissipates with a rising cloud base, and a lesser rising of the cloud top. However, radar observations (Fig. 2b) show a cloud that dissipates with a cloud top that is lowering. After aerosol removal (and temperature nudging is turned off) in the OLI simulation, the entire boundary layer cools and stabilizes (not shown). As a result of this stabilization, turbulence generated by cloud top cooling is not able to extend as far down as before, resulting in a rising cloud bottom. It is not clear what is causing the cloud top to lower in the observed case, but this difference in the cloud shape during dissipation - combined with the much faster observed LWP response compared to simulations - indicates that the observed dissipation is likely due to larger-scale factors such as the possible weak frontal passage described in section 2.1.1. We also speculate that the liquid water profile added to the model initialization results in cloud-top LWC





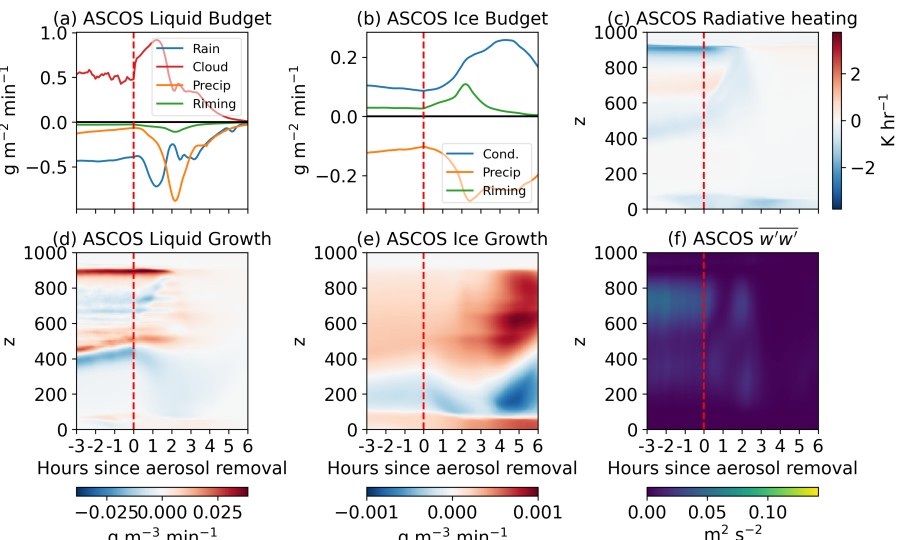

**Figure 8.** Same as Figure 7 for ASCOS.

high enough to produce stronger longwave cooling, which could cause a thermodynamic adjustment that raises the top of the boundary layer.

## 3.2 ASCOS

The simulated LWP response in ASCOS (Fig 6b) is much more in-line with observations than either of the other two simulations. While there is significant variation in observed LWP between 06:30-11:00 UTC, the modeled LWP fits the downwards trend of these variations quite closely. However, the simulations show that the main cloud layer dissipates almost entirely within about hours after aerosol removal and thereafter most of the liquid is contained in a fog layer near the surface (Fig. 5b). This fog layer may be in line with observations; while not detected by radar, observers reported a fog bow forming in the later hours (UTC) of August 31st (Mauritsen et al., 2011). Ice falls slowly out of the atmosphere. The radar observations show that the cloud dissipated with a simultaneous drop in cloud top height. The cause is not clear, but it may be a result of ice slowly settling after the liquid is mostly removed, as seen in our simulations. It may also be associated with a change in the large-scale divergence and subsidence rate. Whatever the cause, the drop in cloud top height (as indicated by the presence of either liquid or ice) in the simulations seems to occur much more rapidly. While the observations and simulation are not exactly the same, they seem similar enough that a lack of aerosol particles cannot be ruled out as a cause of the dissipation in the ASCOS case.

Figure 8 shows the budgets for ASCOS in the same fashion as OLI in Figure 7. The ASCOS and OLI cases are similar, both with constant liquid and ice precipitation reaching the surface throughout the pre-aerosol removal period, balancing the positive ice and liquid growth. Both simulations had uniform radiative cooling at cloud top and weak heating or cooling elsewhere. Unlike OLI, the ASCOS simulation was initialized with a sounding where the boundary layer was decoupled from



the surface from a temperature inversion around 350 m (Fig. 3a). As a result, the vertical turbulent momentum flux (Fig 8f) does not extend as far below the cloud as in OLI (Fig. 7f).

Much like OLI, once aerosol are removed from the environment there is a sharp increase in cloud condensational growth, leading to a large amount of liquid precipitation and rain evaporation. Shortly after this rise in liquid condensation, ice growth rates almost triple at 09:00 UTC. Simultaneously, liquid growth drops sharply. While initially this may seem to indicate the WBF process glaciating a cloud, investigating the growth budget in 2D (Fig. 8(d-e)) shows that there is typically net condensation of liquid in the cloud layer after aerosol removal. However, liquid growth (while remaining positive) is still decreased significantly in the location of maximal ice growth, so it appears that both liquid and ice processes are competing for available water vapor. As liquid droplets and ice crystals grow, collide, and fall out as precipitation, this moisture is removed from the atmosphere and eventually no supersaturation exists with which to grow any hydrometeors. Like OLI, precipitation processes seem to drive the liquid dissipation and the post-removal boundary layer cools throughout (not shown). However, due to the existing decoupled nature of the boundary layer, the effect of this stabilization on the turbulence of the cloud is weakened compared to OLI. As a result, while a slight cloud base rising is observable in, for example, Figure 8c it is not as pronounced as in OLI.

### 3.3 SMT

The SMT simulation is notably different than the other two, with simulated LWP reaching near-zero values within only 2 hours of aerosol removal (Fig. 6c). Observed LWP drops even faster, within 1 hour of the observed drop in aerosol concentration, similar to the observed LWP at Oliktok Point. Observed LWP values are near-zero by 07:00 UTC, but radar returns are detected for the next several hours. Based on images taken from the measurement site and micropulse LIDAR depolarization ratios (not shown), it is likely that the radar returns detected after 07:00 are primarily due to near-surface ice fog and not to the presence of liquid water. LWP values are more variable after 09:00 UTC, but this is most likely due to higher-level liquid clouds passing overhead (which can also be seen on radar in Fig 4b). See Guy et al. (2021) for more analysis of these observations. Radar also shows a cloud whose top is lowering but Figure 5c shows little lowering of cloud top.

Figure 9 shows the liquid and ice budget, radiative heating rate, and vertical turbulent momentum flux for the SMT case. Radiative cooling at cloud top is about twice as strong ($\sim 4$ K h$^{-1}$) as in ASCOS or OLI ($\sim 2$ K h$^{-1}$). This is in agreement with previous studies which show that an increase in aerosol (200 cm$^{-3}$ in SMT versus 80 cm$^{-3}$ and 89 cm$^{-3}$ in OLI and ASCOS, respectively) leads to enhanced maximum cloud top cooling in Arctic clouds (e.g. Williams and Igel, 2021). The SMT sounding used to initialize RAMS (Fig. 4a) was well-mixed from cloud top to near the surface, so turbulent momentum fluxes are able to consistently extend down to about 200 m above the surface. As a result of the stronger cooling at cloud top, vertical momentum flux ($\overline{w'w'}$, Fig, 9f) at SMT is $\sim 0.75$ m$^2$ s$^{-2}$ - approximately 10x stronger than seen in OLI or ASCOS.

Unlike the other two cases, the simulated SMT cloud reaches a different equilibrium pre-aerosol removal. Instead of a balance between cloud droplet/ice growth and precipitation, like in OLI and ASCOS (Figs 7 and 8), the SMT cloud is balanced by cloud droplet growth at cloud top and cloud base but cloud droplet evaporation in the interior and along the top and bottom edges (Fig. 9c. There was no liquid precipitation or rain evaporation prior to aerosol removal.





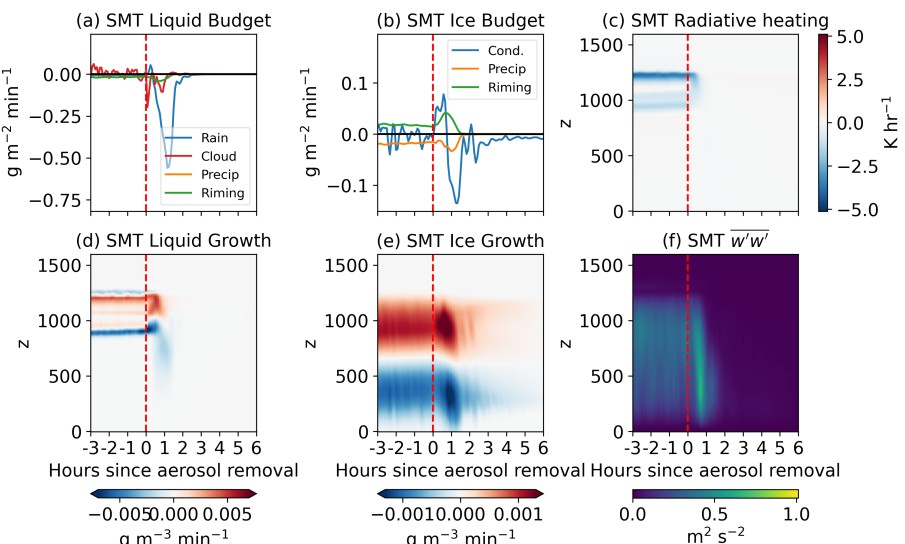

**Figure 9.** Same as Figures 7 and 8 for SMT.

Similarly, column-integrated ice deposition fluctuates around zero prior to aerosol removal unlike in the other two simulated cases (Fig. 9b). While there is significant ice production and growth in-cloud (Fig. 9e), it is balanced out by a near-total sublimation below cloud (Fig. 9e), in part due to the much drier below-cloud boundary layer seen in SMT compared to OLI

and ASCOS (Figs. 2-3). While some ice precipitation did accumulate on the surface (Fig. 9b, orange line), that removal from precipitation mirrors the the addition of ice due to riming, with precipitation being slightly lower due to sublimation of the rimed ice while falling through the below-cloud atmosphere.

Another difference between OLI and ASCOS and the SMT simulation is the location of the ice relative to the liquid layer. In OLI and ASCOS, there was a more distinct separation between the liquid layer and the ice precipitation below it (Fig. 5). In

SMT, by contrast, ice is present throughout the liquid layer. This creates more competition between the liquid and ice phases in the cloud.

After aerosol removal there is a short period of rain/drizzle production, though most droplets evaporate or rime before reaching the surface; this increase in rain is not visible as precipitation but instead as rain evaporation in Figure 9a and riming in 9b. The areas of liquid evaporation shortly after 06:00 UTC at ∼750 m are coincident with strong ice growth (Fig. 9e),

suggesting some role of the WBF process in dissipating the cloud. However, this is not necessarily the case. The enhanced negative liquid growth in Fig. 9d occurs at and below cloud base. Cloud droplets grow in the net by condensation (Fig. 9a, red line) and rain evaporation becomes the largest sink of liquid water (blue line). This indicates that the removal of aerosol is promoting development of rain-sized liquid drops which are falling out but not actually reaching the surface (as there is no surface accumulation visible in Fig. 9a). After a very short amount of time (<1 hour), the ice hydrometeors large enough to

precipitate do so, and both ice and liquid evaporate. This is in contrast to the other two cases, where ice growth was always positive.





SMT also had less initial liquid water than the other two simulations ($\sim$30 g m$^{-2}$ compared to $\sim$100 g m$^{-2}$ in both OLI and ASCOS), which also may explain why this cloud dissipated faster as there was less than half the amount of liquid water that needed to be removed from the atmosphere. Post-removal, the boundary layer cools much like OLI and ASCOS (not shown).

However, in contrast to those two cases, the boundary layer remained well-mixed for the first hour, and becomes more stable around 08:00 UTC - 2 h after aerosol removal and at which point the cloud has mostly dissipated. As a result, turbulent vertical motions extend quite far throughout the boundary layer (Fig. 9f). This combined with the stronger turbulence generated by enhanced cooling at cloud top and the drier atmosphere above and below cloud means that dry air is more readily entrained into the environment, and may have been a reason why dissipation occurred so much more quickly. The increased vertical

momentum flux seen immediately after aerosol removal (Fig. 9f) may be due to drag from precipitating raindrops.

## 4 Discussion and Conclusions

We investigate the role of low aerosol concentrations on the dissipation of Arctic mixed-phase clouds by running LES simulations of three different cases in different locations (Alaska's northern coast [OLI], Arctic ocean ice floe [ASCOS], and Greenland's summit station [SMT]). Each LES simulation is initialized based on observations and aerosol concentrations are

instantaneously forced to 0 cm$^{-3}$ at a specified time. Comparing the simulated LWP to observations (Fig. 6), we are able to determine whether it is *possible* that each observed case dissipated due to a lack of aerosol. By removing all aerosol instantaneously and preventing the model from nucleating new hydrometeors, we effectively simulate the *fastest possible* response to a lack of aerosol. A case that is observed to dissipate faster than its simulation, in this setup, is likely to have other factors driving the dissipation. We find OLI to be one such case. Cases where the observed LWP response is similar to the modeled

response are more likely to be actually caused by a lack of aerosol. The ASCOS and, to some extent, SMT cases fall into this category. While noisy, the observed ASCOS LWP trend lines up very well with the simulation. The SMT case is less certain and while the observed LWP decreases faster than the simulated values, some factors (such as less certain/accurate representativeness of this cloud setup in the model) may explain the difference. We argue that the ASCOS and SMT cases should be investigated further as possible cases of aerosol-limited dissipation. Avenues for further study include simulations with more

realistic aerosol treatment, exploring the relative impact of above and below cloud aerosol on dissipation, and investigating the role of IN entrainment

Our simulations revealed two pre-aerosol removal equilibrium balance states which respond differently to aerosol removal. The first, seen in OLI and ASCOS, results in a continually precipitating cloud in both ice and liquid where droplet growth is balanced by loss through collisions resulting in removal by precipitation. Conversely, the second balance state (seen in SMT)

occurred in a thinner, colder cloud with no liquid precipitation and very little ice precipitation. Instead, liquid and ice coexist in a larger area in SMT (Fig. 5), creating more competition for the available water vapor. The stronger turbulent motions caused by enhanced cloud top cooling in SMT caused the relatively drier above-cloud air to be mixed and entrained into the cloud, also prohibiting the development of large enough supersaturations needed for constant growth.

In addition, we found little difference in microphysical response to aerosol removal between a coupled boundary layer (OLI)
and a decoupled one (ASCOS). SMT and ASCOS both had decoupled boundary layers, and SMT showed a very different
response to aerosol removal than both OLI or ASCOS. We believe that, given the evidence from these three simulations, the
microphysical balance state of the cloud is more important to determining the response to aerosol removal than boundary layer
properties.

Understanding Arctic energy balances are paramount to studying the Earth's climate as a whole. Low-level mixed phase
clouds have been shown be a large regulator on the Arctic climate, and understanding these clouds and their processes is
important to furthering our understanding and modeling ability of weather and climate. While we believe aerosol-limited
dissipation in the Arctic to be an uncommon (if not rare) event, understanding the impact of a pristine Arctic environment and
how it might change in a more polluted future will be necessary steps in researching Earth's climate change and its impacts.

*Code and data availability.* Model source code is available at https://github.com/lsterzinger/arctic-rams-6.1.22. Horizontally averaged model
data from RAMS is available at https://doi.org/10.5281/zenodo.5851364. Full 3D model data can be obtained by emailing the author at
lsterzinger@ucdavis.edu. Fully reproducible code for all figures is available at https://doi.org/10.5281/zenodo.5851414

*Author contributions.* LS, AI, and JS conceptualized the study. LS and AI designed and performed the simulations. All authors contributed
to data analysis. LS wrote the manuscript with input from all authors.

*Competing interests.* The authors declare that they have no competing interests.

*Acknowledgements.* This work was supported by United States Department of Energy Atmospheric System Research Grant #DE-SC0019073-
0. ICECAPS-ACE was funded by NSFGEO-NERC grant 1801477. We would like to thank the technicians at Summit Station and the science
support provided by Polar Field Services whose efforts were crucial to maintaining data quality and continuity at Summit.





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
