# Peer review of "Do arctic mixed-phase clouds sometimes dissipate due to insufficient aerosol? Evidence from comparisons between observations and idealized simulations"

_Atmospheric Chemistry and Physics, 2022_

## Referee Comment (RC1)

**Review** of manuscript **acp-2022-36**: *Arctic mixed-phase clouds sometimes dissipate due to insufficient aerosol: evidence from observations and idealized simulations* by Sterzinger, L. J., Sedlar, J., Guy, H., Neely III, R. R., and Igel, A. L.

**Overview**

This paper investigates the impact of a sudden decrease of aerosol concentrations on the dissipation of mixed-phase clouds in the Arctic boundary layer. The importance of understanding low-levels clouds in the Arctic has been well established, as well as the fact that lot of this understanding is still lacking. One step in improving scientific knowledge of Arctic mixed-phase clouds can be this paper. It starts by a brief recapitulation of the problem, pick a certain niche (single-layer mixed-phase clouds during Arctic spring and summer), and chooses three representative examples to motivate a novel model study. The model study uses well established software, RAMS, to conduct idealized large-eddy simulations. The results are then examined, and a clear conclusions are drawn. While this paper has a high potential, it has also some significant shortcomings:

1. Not distinguishing between causation and correlation in the observations;

2. Lack of replicability due to incomplete description of the methodology;

3. Omitting discussion of dynamic effects.

Each of these major issues is described in its own section below. I believe that authors will be easily able to fix them.

**Overall recommendation**

Publish after major revisions

**Major Comment 1**

The causational relationship between the decrease in the aerosol concentrations and changes in changes in the properties of the clouds are expected. However, this paper goes an extra steps and claims that observed dissipation of clouds is most likely due to decrease in the aerosol concentrations. However, observations as described in the section 2.1 show only the correlation of these two events. Furthermore, figure 2.b and 2.c show that the cloud started dissipating while the aerosol concentrations were still high (please see the red line in the snapshot below).

[Figure]

Therefore, it is possible that the the causation could work the other way around: increased precipitation could possibly scavenge aerosols from the lower part of the boundary layer. Or it is also possible that both changes are caused by an advection of slightly different air mass. Unfortunately, I do not see this addressed discussed in the paper.

Furthermore, it leads to slightly misleading part of the title: "evidence from observations and idealized simulations". What we currently see in the paper is "evidence from idealized simulations motivated by observations."

**Major Comment 2**

The section "2.3 Experimental setup" is well written, yet clearly incomplete, which means that the study is in the current form not replicable.

The following properties of the setup of simulations are missing:

- radiation scheme: which radiation scheme is used? Is it coupled to the microphysics?

- surface conditions: considering that most of the simulations are during the day, the surface albedo might play a role. But its setting is not described.

- wind velocity: what is the initial profile of wind velocity, and is it derived from observations, or the reanalysis?

- meso-scale forcing: is the initial wind velocity maintained?

- upper boundary conditions: Do you use the *Gravity wave radiation condition (Klemp and Durran, 1983)*, or something else?

Speaking further about the replicability, I was not able to find the case setups in any of the online repositories linked in the data statement. The case setup files are either missing, or are well hidden in somewhere deep within the directory tree. I would like to ask the authors to fix that as well before the submission of the review manuscript.

**Major Comment 3**

The paper omits discussing the dynamic effects of the boundary layer, and whether they are represented in the model.

- Dynamic effect in general: the cloud dissipation in OLI and SMT could be affected by wind shear driven entrainment. Unfortunately the data in the online repository do not show the wind velocity data, and the simulation setup is unclear (see last paragraph of Major Comment 2).

- Dynamic effect in general II: How much does the increased precipitation effect the surface fluxes? (see Major Comment 2)

- Dynamic in the model: The *temperature nudging* is usually applied above the boundary layer. However here it is applied withing the boundary layer, which could be a serious issue. The whole cloud dissipation could be caused by the sudden removal of the cloud. Have you performed a sensitivity test for that? Have you also checked how it affects model spin-up?

- Dynamic limitations of the model: When the clouds disappear and some of the model levels remain supersaturated, is the dynamic core of the simulation still working correctly?

**Minor Comments**

line 99: ”ASL“
The abbreviation ASL should be defined. The default meaning of ASL is ”Atmospheric Sciences Laboratory” or ”American Sign Language”.

Figure 2:
The panels b, c, and d are very small. Expanding them the figure on the full width of the page would help.

Figure 2, panel c:

- missing label of the axis y.

- related to the point above, is there a specific meaning for ”z“ in *00z, 03z, 09z*, or is it just a formatting issue?

Figure 2, panel d:
Adding longitude and latitude to axis would be nice, or at least adding some other geographical

coordinates.

Line 109: "(CPC) (measuring particles 10-3000 µm"
Did the instrument only measure particles larger than 10 micrometer? That would mean that wast majority of aerosols was not recorded. Or do you mean "nm"?

Figure 3:
Same issues as Figure 2

Figure 3.a:
The potential temperature profile is missing, only relative humidity is shown. Please add it to the plot.

Figure 4:
Same issues as Figure 2

Line 153: there is a dry layer at 400 m.
The panel 3.a shows only a very minor decrease in humidity.

Line 177: "... aggregates, snow, hail, and graupel"
Are aggregates a separate category from snow?

Lines 178–179:
The list of the microphysical processes is slightly confusing.

- It seems that *sublimation* is missing

Figure 5:
Considering that three cases are compared with respect how fast the clouds dissipate, it would make more sense to show figures under each other.

Line 237–238: "all further discussion will be discussed"
It would be better to write "all features will be further discussed..." or "all events will be further discussed"

Figure 7, panel a:
The legend is confusing. The caption of the figure does not explain any of the terms there. Therefore reader can't know

- what is the difference between "Rain" and "Precip",

- what is the line "Cloud".

Figure 7, panel b:
The legend is confusing here as well. The caption of the figure does not explain any of the terms. It seems that deposition is not considered in the ice budget and instead "Cond" is.

Figure 8:
Same as figure 7

Figure 9:
Same as figure 7

Line 276: "about hours after"
How many hours?

Line 276: "in a fog layer near the surface"
This does not make sense. How could there form a fog layer when all aerosols were removed?

Line 281: "b y"
by

Line 293: growth budget in 2D (Fig. 8(d-e))
Does 2D refer to height and time?

Line 293: "different than the other two"
More fitting would be "different from"

Line 293: "...in SMT caused the relatively drier above-cloud air to be mixed and entrained into the cloud,"
This is a good point, but it is not shown in the result part.

Line 376–379: "We believe that, given the evidence from these three simulations, the microphysical balance state of the cloud is more important to determining the response to aerosol removal than boundary layer properties"
This is a very strong statement, and currently not supported by results.
The effect of other boundary layer properties (such as wind shear or surface forcing) is not compared in the results. This seems more like a proposal for a future research.

Line 384: 6.1.22.
Insert a space between 22 and dot, so the link will work.

Lines 400–401:
The title of the paper does not have to been in capitals.

Line 530: Sterzinger, L.: Data for Sterzinger et al. (2022, in Prep)
This should be in the data statement only, not in the References.

Line 532: Sterzinger, L.: Plotting Scripts for Sterzinger et al (2022) (in Prep),
This should be in the data statement only, not in the References.

---

## Referee Comment (RC2)

**Review for acp-2022-36:**

**_'Arctic mixed-phase clouds sometimes dissipate due to insufficient aerosol: evidence from observations and idealized simulations'_**

**by Sterzinger et al.**

Arctic clouds remain a great challenge for the climate and weather forecast community, as the processes that determine their life-cycle are poorly understood until today. This study uses a Large-Eddy Simulation to investigate observed cases of rapidly dissipating clouds. The authors explore the hypothesis that a limited aerosol availability may be the primary cause for the observed cloud depletion. This is a very interesting and well-written paper. I have only one major concern regarding the experimental set-up and its realism (see main comment 1). Apart from this, addressing my comments below should not be very time-consuming, thus my recommendation is minor revision.

**Main comments:**

**1)** The authors explain the methodology between lines 200-205. However this paragraph concerns the aerosols that drive CCN activation. Are only CCN removed in the sensitivity simulations? Do INs remain unaffected? If only CCN are modified, I wonder to which extent this can be realistic. If e.g. aerosol transport changes drastically due to changing large-scale conditions, shouldn't this affect both CCN and INP availability (especially in decoupled environments were surface aerosol sources are expected to have limited impact)? If all aerosols are removed ($n_{aer}$ and $n_{INP}$) please state this explicitly in the text. If not, it is worth performing additional simulations with no CCN/INPs at all. While a lack of CCN leads to decreasing cloud liquid, decreasing INP concentrations can reduce the efficiency of WBF process and change the timescales for cloud dissipation.

**2)** I think that the impact of boundary layer stability is not discussed as much, while it can be very important. For example, while OLI and ASCOS cases are discussed as similar in the text, the ASCOS momentum flux seems about a factor of two weaker than the OLI flux. It is worth investigating and discussing in more detail how the initial thermodynamic state affects cloud evolution. Also if high time-resolution thermodynamic profiles are available (e.g. from radiometers), it should be investigated whether the changes in

modelled thermodynamic stability after aerosol depletion conform with observations. If significant deviations are found between model and observations, this might explain to some extent the deviations in LWP evolution (Figure 6).

**3)** the authors state in the abstract that cloud response to rapid aerosol depletion is case-dependent. However (following my previous comment) is it possible to draw any conclusion regarding the thermodynamic/macrophysical conditions that are more likely to lead to cloud dissipation in the absence of significant aerosol forcing?

**Minor Comments:**

**Line 187:** $n$ in DeMott formula represents aerosol concentration at STP conditions ($scm^{-3}$)

**Section 2.3:** I would appreciate more information on the experimental set-up. What profiles are used to force the model? Is it only the potential and RH profiles shown in the figures? Also what about the surface conditions? Is the model run with fixed surface temperature or fixed surface fluxes? What about the assumed surface roughness and albedo? Or is there a surface model? If fixed surface values are used, then state the actual numbers. How long is the spin-up time?

**Line 205:** For how long is nudging applied (6 hours as indicated in the plots?)? Also why so strong nudging in the PBL is necessary in a model that does not account for varying large-scale forcing? What happens if you don't apply nudging before aerosol removal?

**Lines 265-270:** Could erroneous cloud top displacements in the model be corrected with a better constrained large-scale subsidence? It wouldn't be strange if the same horizontal divergence is not suitable for all three cases.

---

## Author Comment (AC1)

**Response to RC1**

We thank the reviewer for their time and thoughtful feedback. Both reviews pointed out that more information was needed in the experimental setup section (sec. 2.3) and as such we have added more info and clarified details on the experiment setup that make the manuscript more easily understandable. Detailed responses to individual comments are posted below. A track-changes copy of the revised manuscript is provided at the end of this document.

**Major Comment 1**

The causational relationship between the decrease in the aerosol concentrations and changes in changes in the properties of the clouds are expected. However, this paper goes an extra steps and claims that observed dissipation of clouds is most likely due to decrease in the aerosol concentrations. However, observations as described in the section 2.1 show only the correlation of these two events. Furthermore, figure 2.b and 2.c show that the cloud started dissipating while the aerosol concentrations were still high (please see the red line in the snapshot below).

[Figure]

Therefore, it is possible that the the causation could work the other way around: increased precipitation could possibly scavenge aerosols from the lower part of the boundary layer. Or it is also possible that both changes are caused by an advection of slightly different air mass. Unfortunately, I do not see this addressed discussed in the paper.

> The reviewer is correct in their assessment of the interchangeability of correlation. It was not our goal to state with any degree certainty that any of the cases were the result of aerosol-limited dissipation, only to suggest that it was a possible mechanism. We did briefly mention a possible change of airmass (initial submission; lines 131-133) and that the OLI case specifically does not appear to be an example of aerosol limited dissipation (line 240) but the reviewer is correct that more attention needs to be given to alternate possible methods of dissipation that would result in the aerosol/radar signals we presented (such as the reviewer's suggestion of wet scavenging). We have made this more explicit in the manuscript such as in lines 127-136, 274, 299-302,

Furthermore, it leads to slightly misleading part of the title: "evidence from observations and idealized simulations". What we currently see in the paper is "evidence from idealized simulations motivated by observations."

> The title has been changed to more accurately reflect the content of the paper to "Do Arctic mixed-phase clouds sometimes dissipate due to insufficient aerosol? Evidence from comparisons between observations and idealized simulations."

**Major Comment 2**

The section "2.3 Experimental setup" is well written, yet clearly incomplete, which means that the study is in the current form not replicable.

The following properties of the setup of simulations are missing:

- radiation scheme: which radiation scheme is used? Is it coupled to the microphysics?
- surface conditions: considering that most of the simulations are during the day, the surface albedo might play a role. But its setting is not described.
- wind velocity: what is the initial profile of wind velocity, and is it derived from observations, or the reanalysis?
- meso-scale forcing: is the initial wind velocity maintained?
- upper boundary conditions: Do you use the Gravity wave radiation condition (Klemp and Durran, 1983), or something else?

> This section does indeed require more detail and has been expanded to address the points from both reviewers. Section 2.3 has been expanded with more detail on the modeling setup used in this study.
>
> Answering the specifics questions addressed here: The radiation scheme used is based on Harrington (1997), and accounts for all 7 hydrometeor types represented in the model. The surface type was set as ice and surface fluxes were turned off (since they are

expected to be near zero over ice, see e.g. Shupe et al. 2013a). There is no mesoscale forcing (or forcing of any kind, other than the temperature nudging). Periodic boundary conditions were employed, with the initial wind field provided by the radiosonde used to also initialize the temperature, pressure, and moisture profiles. The vertical boundary condition uses a Rayleigh friction absorbing layer which relaxes all three velocity components and potential temperature to their horizontally homogeneous reference state values.

Speaking further about the replicability, I was not able to find the case setups in any of the online repositories linked in the data statement. The case setup files are either missing, or are well hidden in somewhere deep within the directory tree. I would like to ask the authors to fix that as well before the submission of the review manuscript.

This was an oversight on our part. The RAMS namelists and initial sounding for each case have been added to the data repository at https://doi.org/10.5281/zenodo.6502720. Note that the ASCOS sounding is provided directly within the RAMSIN (namelist) file, whereas the soundings for OLI and SMT are provided in separate SOUND_IN files.

**Major Comment 3**

The paper omits discussing the dynamic effects of the boundary layer, and whether they are represented in the model.

- Dynamic effect in general: the cloud dissipation in OLI and SMT could be affected by wind shear driven entrainment. Unfortunately the data in the online repository do not show the wind velocity data, and the simulation setup is unclear (see last paragraph of Major Comment 2).

  The model output in the repository contain U/V/W velocities, as do the soundings used to initialize the model. For clarity, we have added wind barbs to the soundings in plots 2-4. These sounding were used to initialize the model, and minimal wind shear is seen in OLI and SMT, with directional shear above the cloud layer observed in ASCOS (though wind speeds are relatively weak, approximately 5-10 knots.

- Dynamic effect in general II: How much does the increased precipitation effect the surface fluxes? (see Major Comment 2)

  Surface fluxes are disabled in the model (see response to Major Comment 2)

- Dynamic in the model: The temperature nudging is usually applied above the boundary layer. However here it is applied withing the boundary layer, which could be a serious issue. The whole cloud dissipation could be caused by the sudden removal of the cloud. Have you performed a sensitivity test for that? Have you also checked how it affects model spin-up?

With the lack of large-scale forcing (other than subsidence), the temperature nudging needed to be applied throughout the boundary layer as without it the cloud would slowly dissipate over the course of a few hours. In order to keep the cloud at near-observed conditions while the model spins up, the nudging was applied until aerosol were removed. The cloud dissipation that occurs without nudging is a slow enough process that we do not believe that keeping the nudging active until aerosol are removed has any significant effect on the results, as the dissipation due to the lack of CCN acts on a much faster timescale. Below is a figure of LWP in the ASCOS with temperature nudging turned on (as presented in the manuscript) in the blue line and with temperature nudging turned off throughout the simulation in orange. The response of LWP to aerosol removal occurs at near-identical rates between the two simulations, indicating that the nudging scheme had little effect on cloud dissipation post-removal.

[Figure]

- Dynamic limitations of the model: When the clouds disappear and some of the model levels remain supersaturated, is the dynamic core of the simulation still working correctly?

  RAMS prognoses total water content (condensed water + vapor) as well as condensed water for each hydrometeor species. Water vapor (and supersaturation) is a diagnosed quantity. We have no reason to believe that the dynamic core is working improperly.

**Minor Comments**

line 99: "ASL"
The abbreviation ASL should be defined. The default meaning of ASL is "Atmospheric Sciences Laboratory" or "American Sign Language".

> We meant "above sea level". There is no specific guideline for this abbreviation in the ACP guidelines, but we changed ASL to the more standard format of "m a.s.l." (meters above sea level). We have also defined this acronym in the text when it first appears on line 99.

Figure 2:
The panels b, c, and d are very small. Expanding them the figure on the full width of the page would help.

> We have shifted a few panels on this figure, which should make it more legible. We have also increased the figure width to span the entire page.

Figure 2, panel c:

- missing label of the axis y.

  > We have added a label of "Number [cm^-3]" to the y-axis
- related to the point above, is there a specific meaning for "z" in 00z, 03z, 09z, or is it just a formatting issue?

  > "z" is commonly used in meteorology and aviation to denote UTC ("zulu") time in the USA. We have changed removed the "z" and changed the axis label to "Hour (UTC)" for clarity

Figure 2, panel d:
Adding longitude and latitude to axis would be nice, or at least adding some other geographical coordinates.

> Lat/lons have been added to these figures

Line 109: "(CPC) (measuring particles 10-3000 μm"
Did the instrument only measure particles larger than 10 micrometer? That would mean that wast majority of aerosols was not recorded. Or do you mean "nm"?

> The reviewer is correct – this is a typo and has been changed to nanometers

Figure 3:
Same issues as Figure 2

> See response to figure 2 comments

Figure 3.a:
The potential temperature profile is missing, only relative humidity is shown. Please add it to the plot.

> We apologize for leaving out this information. This was a mistake and has been corrected.

Figure 4:
Same issues as Figure 2

> See response to figure 2 comments

Line 153: there is a dry layer at 400 m.
The panel 3.a shows only a very minor decrease in humidity.

> This line has been removed

Line 177: "... aggregates, snow, hail, and graupel" Are aggregates a separate category from snow?

> Yes, the RAMS model has a separate category for aggregate snowflakes

Lines 178–179:
The list of the microphysical processes is slightly confusing.

- It seems that sublimation is missing

> This list has been reworded for better understandability

Figure 5:
Considering that three cases are compared with respect how fast the clouds dissipate, it would make more sense to show figures under each other.

> The subplots have been reorganized vertically

Line 237–238: "all further discussion will be discussed"
It would be better to write "all features will be further discussed..." or "all events will be further discussed"

> This sentence has been reworded for clarity

Figure 7, panel a:
The legend is confusing. The caption of the figure does not explain any of the terms there. Therefore reader can't know

- what is the difference between "Rain" and "Precip",

- what is the line "Cloud".

    The figure captions for figures 7, 8, and 9 have been clarified to better define the plotted lines

Figure 7, panel b:
The legend is confusing here as well. The caption of the figure does not explain any of the terms. It seems that deposition is not considered in the ice budget and instead "Cond" is.

    See response to comment on figure 7, panel a

Figure 8:
Same as figure 7

    See response to comment on figure 7, panel a

Figure 9:
Same as figure 7

    See response to comment on figure 7, panel a

Line 276: "about hours after" How many hours?

    This typo has been fixed; it should have read "about three hours after"

Line 276: "in a fog layer near the surface"
This does not make sense. How could there form a fog layer when all aerosols were removed?

    The fog layer formed shortly before aerosol were removed from the model (Fig 5b)

Line 281: "b y" by

    Typo has been fixed

Line 293: growth budget in 2D (Fig. 8(d-e)) Does 2D refer to height and time?

    Yes, this has been clarified in the text (line 327) and in the figure caption (Figs 7-9)

Line 293: "different than the other two" More fitting would be "different from"

    Fixed as requested

Line 293: "...in SMT caused the relatively drier above-cloud air to be mixed and entrained into the cloud,"
This is a good point, but it is not shown in the result part.

We changed this line to "… in SMT may have caused …"

Line 376–379: "We believe that, given the evidence from these three simulations, the microphysical balance state of the cloud is more important to determining the response to aerosol removal than boundary layer properties"
This is a very strong statement, and currently not supported by results.

The effect of other boundary layer properties (such as wind shear or surface forcing) is not compared in the results. This seems more like a proposal for a future research.

This line has been changed to more clearly represent it as a proposal for future work

Line 384: 6.1.22.
Insert a space between 22 and dot, so the link will work.

This link has been updated and fixed

Lines 400–401:
The title of the paper does not have to been in capitals.

This was an error with the journal's citation export utility and has been corrected

Line 530: Sterzinger, L.: Data for Sterzinger et al. (2022, in Prep) This should be in the data statement only, not in the References.

We are following the ACP Submission Guide, "Prepare your Assets" section (https://www.atmospheric-chemistry-and-physics.net/submission.html#assets). The "Data Sets" subsection states that these data citations also be included in the reference list:

"Authors are requested to follow our data policy including

- the deposit of research data (i.e. the material necessary to validate the research findings) that correspond to manuscripts, preprints, or journal articles in reliable FAIR-aligned data repositories that assign persistent identifiers (preferably digital object identifiers (DOIs)). Suitable repositories can be found at https://www.re3data.org/;
- the proper citation of data sets in the text and the reference list including the persistent identifier. For data sets hosted on GitHub, authors are kindly asked to issue a DOI through Zenodo and include this DOI in the reference list;"

Line 532: Sterzinger, L.: Plotting Scripts for Sterzinger et al (2022) (in Prep), This should be in the data statement only, not in the References.

The "Software and Model Code" subsection in the Submission Guide linked above states that these citations are also requested to be in the reference list:

"Authors are encouraged to deposit software, algorithms, and model code in FAIR-aligned repositories/archives whenever possible. These research outputs are then cited in the manuscript using the received DOI and included in the reference list."

[revised manuscript text omitted]

---

## Author Comment (AC2)

**Response to RC2**

We thank the reviewer for their time and thoughtful feedback. Both reviews pointed out that more information was needed in the experimental setup section (sec. 2.3) and as such we have added more info and clarified details on the experiment setup that make the manuscript more easily understandable. Detailed responses to individual comments are posted below. A track-changes copy of the revised manuscript is provided at the end of this document.

**1)** The authors explain the methodology between lines 200-205. However this paragraph concerns the aerosols that drive CCN activation. Are only CCN removed in the sensitivity simulations? Do INs remain unaffected? If only CCN are modified, I wonder to which extent this can be realistic. If e.g. aerosol transport changes drastically due to changing large-scale conditions, shouldn't this affect both CCN and INP availability (especially in decoupled environments were surface aerosol sources are expected to have limited impact)? If all aerosols are removed ($n_{aer}$ and $n_{INP}$) please state this explicitly in the text. If not, it is worth performing additional simulations with no CCN/INPs at all. While a lack of CCN leads to decreasing cloud liquid, decreasing INP concentrations can reduce the efficiency of WBF process and change the timescales for cloud dissipation.

> We agree with the reviewer that this section is lacking in detail and is generally unclear, and we have expanded on the methodology in the revised manuscript. In our prescribed aerosol concentration scheme, while CCN are removed from the model INPs are not. Because of the simplified ice nucleation treatment in the model, INPs are not explicitly represented. The equivalent of removing INPs would be to turn off all ice nucleation. However, immersion freezing of INP already contained within liquid droplets has been shown to be the primary ice nucleation mode in arctic mixed-phase clouds (e.g. Savre and Ekman 2015, https://doi.org/10.1002/2014JD023000 ). In the context of large-scale aerosol decrease that we are examining in this study, these immersed INP would persist beyond the "aerosol removal time". As such, we had decided to maintain ice nucleation after the aerosol removal time.

> However, in light of your comment, we performed simulations in which we removed CCN and turned off ice nucleation. LWP was affected, but for SMT and ASCOS the LWP response was similar to the baseline simulations – just delayed slightly. OLI showed a ~50% increase in the time required to decrease to near-zero, but post-removal liquid budgets were qualitatively similar. The LWP evolution for these new simulations is now shown in the revised Figure 6 (see below) and discussed in the text. Since the choice of ice treatment post-removal does not qualitatively impact our results, we decided to keep the analysis of the original simulations in the remainder of the manuscript.

[Figure]

**2)** I think that the impact of boundary layer stability is not discussed as much, while it can be very important. For example, while OLI and ASCOS cases are discussed as similar in the text, the ASCOS momentum flux seems about a factor of two weaker than the OLI flux. It is worth investigating and discussing in more detail how the initial thermodynamic state affects cloud evolution. Also if high time-resolution thermodynamic profiles are available (e.g. from radiometers), it should be investigated whether the changes in modelled thermodynamic stability after aerosol depletion conform with observations. If significant deviations are found between model and observations, this might explain to some extent the deviations in LWP evolution (Figure 6).

> Unfortunately, high time-resolution thermodynamic profiles are not available, so direct comparison of boundary layer evolution between observations and the model is not possible.

> We apologize that the ASCOS sounding was not present in the submitted manuscript, this has been corrected. The ASCOS case had a stable layer near the surface, whereas the OLI case was well-mixed throughout. This, coupled with the higher radiative cooling rates seen in the OLI simulation, explains why OLI was able to sustain momentum fluxes 2x stronger than in ASCOS and certainly points to the importance of the initial thermodynamic state. Doing a systematic study of the influence of the initial thermodynamic state is beyond the scope of this study.

**3)** the authors state in the abstract that cloud response to rapid aerosol depletion is case-dependent. However (following my previous comment) is it possible to draw any conclusion regarding the thermodynamic/macrophysical conditions that are more likely to lead to cloud dissipation in the absence of significant aerosol forcing?

> In-depth investigation on the thermodynamic/macrophysics conditions needed to cause dissipation outside significant aerosol forcing are outside the scope of the current study, though certainly they are an important factor to consider. We have made it more explicit in the conclusions that more work with different boundary layer setups, surface fluxes, and large-scale forcings is needed to fully understand the relative effects of dynamics and microphysics in these cases

**Minor Comments:**
**Line 187:** $n$ in DeMott formula represents aerosol concentration at STP conditions ($scm^{-3}$)

> The units for aerosol concentration have been added (line 196)

**Section 2.3:** I would appreciate more information on the experimental set-up. What profiles are used to force the model? Is it only the potential and RH profiles shown in the figures? Also what about the surface conditions? Is the model run with fixed surface temperature or fixed surface fluxes? What about the assumed surface roughness and albedo? Or is there a surface model? If fixed surface values are used, then state the actual numbers. How long is the spin-up time?

> More detailed information on model configuration has been added to section 2.3. To answer the reviewer's specific comments: The profiles shown in Figures 2-4 were used to initialize each simulation with potential temperature, vapor mixing ratio, and wind data.

> The surface temperature and moisture was set to be equal to the lowest atmosphere level, turning off surface fluxes. All simulations were allowed to spin-up for 6 hours prior to aerosol removal.

**Line 205:** For how long is nudging applied (6 hours as indicated in the plots?)? Also why so strong nudging in the PBL is necessary in a model that does not account for varying large-scale forcing? What happens if you don't apply nudging before aerosol removal?

> The temperature nudging is applied from model initialization until the time at which aerosol are removed (6 hours later in all simulations). Without the nudging, and in the absence of any other large-scale forcing in the model, the cloud would slowly dissipate. The nudging was employed to allow the model time to spin-up while still maintaining a stable cloud. This has been clarified in the text (223-230).

**Lines 265-270:** Could erroneous cloud top displacements in the model be corrected with a better constrained large-scale subsidence? It wouldn't be strange if the same horizontal divergence is not suitable for all three cases.

> A better constrained large-scale subsidence rate certainly has the potential to improve the representation of clouds in the model. Temporal changes in subsidence rates are a potential reason for dissipation in the first place. While we have not run new simulations with a different subsidence rates, we now discuss the possible role of subsidence in the main text.

[revised manuscript text omitted]

---

## Referee Report (RR1)

**Review** of manuscript **acp-2022-36**: *Do arctic mixed-phase clouds sometimes dissipate due to insufficient aerosol? Evidence from comparisons between observations and idealized simulations* by Sterzinger, L. J., Sedlar, J., Guy, H., Neely III, R. R., and Igel, A. L.

**Overview**

Overall, this revised manuscript shows a clear improvement from the initial submission. The methodology of the study is now explained sufficiently and the results are now presented in a better way. That said, there are significant "disagreements" between what is described in the revised manuscript (section 2.3) and what is the setup of the model study (doi.org/10.5281/zenodo.6514322). Furthermore, chosen methods lead to certain limitations of this study, but this is currently not reflected in the conclusions.

Therefore, in the current version of the manuscript, some parts of methods are misleading. Authors are kindly requested to correct the description of their methodology and briefly clarify the limitations of their study in the conclusions.

**1 Overall recommendation**

Publish after minor revisions

**2 Minor Comments**

**2.1 CCN concentrations**

The description of the CCN concentrations in the manuscript differs from the prescribed subsidence in the model setup files (repository https://doi.org/10.5281/zenodo.6514322).

Table 1:
The initial concentration of CCN in the SMT case is listed ad 200 cm$^{-3}$.

Meanwhile, model setup (SMT-RAMSIN) lists the same CCN concentration as in the OLI case.

```
CCN_MAX = 80., !  CCN (#/mg) normally 170
```

This value also appears in the horizontally averaged concentrations in the model output.

It is of course understandable that authors used similar CCN concentrations in all three cases. However, this has to be clearly stated and also briefly justified.

**2.2 Subsidence**

The description of the subsidence in the manuscript differs from the prescribed subsidence in the model setup files (repository https://doi.org/10.5281/zenodo.6514322).

lines 231–233: "..Large-scale subsidence is applied throughout the simulation by imposing a horizontal divergence of $2 \times 10^{-6} \text{s}^{-1}$ at every model level, with a boundary condition of $w_{sub} = 0$ at the surface."

Meanwhile, model setup (ASCOC-RAMSIN, OLI-RAMSIN, and SMT-RAMSIN) do not prescribe this subsidence, but rather large-scale convergence:

```
CDIVMAX = -6.1224e-6, !  Divergence amplitude (s -1) (negative = convergence)
```

This large scale convergence also appears in the horizontally averaged vertical velocity in the model output.

This is still and interesting example of an idealized model setup. Nevertheless, the description in the methodology has to be corrected.

**2.3   Description of Albedo**

The Experiment Setup (2.3) does not describe the surface albedo. While some of the cases take place during the polar day, the surface albedo is an important property in the radiative scheme.

The model setup prescribed albedo value 0.5 (ASCOC-RAMSIN), respectively 0.6 (SMT-RAMSIN).

These albedo values should be described in 2.3 and justified.

**2.4   Comparison Figure not Consistent**

The figure 5 shows an important comparison of all three cases. Unfortunately the time point of CCN removal differ across it figures. The time scales differs as well.

This figure would be much more useful if the time points and time scales were synchronised across all three panels (in a similar way as it is in figure 6).

This figure would furthermore benefit from being larger in size.

**2.5   Turbulent Decay to Non-LES Regime**

In some of the simulations, the dissipation of the clouds leads to decay of turbulence.

This could be clearly seen in the horizontally averaged variable TKEP in the model outputs. By the end of one of the simulations, it drops to values 0.0005–0.0009 $\text{m}^2/\text{s}^2$, which is almost the same value as in the non-turbulent free atmosphere above the boundary layer (0.0005). This indicates that the subgrid scheme no longer approximated the turbulent cascade and the model no longer resolves large eddies.

While this is totally acceptable result, it would be fair to state in the Discussion and Conclusions that the LES by the end of the simulation does not operate in LES regime, but rather as a fine-scale convection-permitting model.

**2.6 Coupled Boundary Layer**

Line 408:

"...coupled boundary layer (OLI)..."

This statement is in a direct disagreement with the description on line 204:

"...surface fluxes are disabled..."

Perhaps it would make sense to correct line 408 to "previously coupled" or "recently decoupled".

**2.7 Strong Conclusion Statement**

Line 422–424:

"We believe that, given the evidence from these three simulations, the microphysical balance state of the cloud may be more important to determining the response to aerosol removal than boundary layer properties"

The conclusions should clarify for which group of boundary layers this applies (decoupled, & polar day, &, large scale convergence, & weak windshear, etc.).

**2.8–12 Further Minor Comments**

- line 238
  The word "decase" does not seem to fit the context here.

- Figure 7, panels a and b
  The timeseries "precip" differs between the panels.
  Either there is a numerical mistake, or the captions is incorrect in the sense that "precip" refers to two different variables.

- Figure 8, panels a and b
  Same as for Figure 7a,b.

- The figures 7, 8, and 9 show very important results, but the panels are very small. Please expand them.

- Code and data availability.
  The links should be updated to the newest version of repositories.

---

## Author Response (AR2)

**Reviewer #1**

The authors have addressed all comments and have done a great job in documenting the limitations in their model set-up. Since these, however, are not mentioned in the discussion session, I think it would be good to include the term 'idealized' or at least 'semi-idealized' for the description of their LES simulations in line 398.

> We have clarified that these simulations are idealized in the conclusions as suggested by the reviewer.

I also think that the statement in lines 405-406 is slightly strong, so I suggest replacing 'more likely' with 'likely' or 'possibly'. You see small errors in processes like rain autoconversion and WBF can rapidly deplete cloud liquid. For example with this highly idealized aerosol forcing, you achieve an enhanced condensational growth of the existing droplets. Thus the diameter threshold for droplet-to-rain autconversion can likely be reached much faster than in the case of a more gradual aerosol reduction.

> We have clarified in this section that we believe that our simulations (ASCOS and SMT) that agree more closely with observations are more likely to have dissipated from a lack of aerosol than OLI, which did not. Of course, just because the LWP response is similar between model and observations does not mean that the causality behind dissipation must be the same. We have clarified in this section that model-obs agreement in LWP decrease suggests to us that those cases should be investigated further.

**Reviewer #2**

**Overview**

Overall, this revised manuscript shows a clear improvement from the initial submission. The methodology of the study is now explained sufficiently and the results are now presented in a better way. That said, there are significant "disagreements" between what is described in the revised manuscript (section 2.3) and what is the setup of the model study (doi.org/10.5281/zenodo.6514322). Further- more, chosen methods lead to certain limitations of this study, but this is currently not reflected in the conclusions.

Therefore, in the current version of the manuscript, some parts of methods are misleading. Authors are kindly requested to correct the description of their methodology and briefly clarify the limitations of their study in the conclusions.

**1 Overall recommendation**

Publish after minor revisions

**2 Minor Comments**

**2.1 CCN concentrations**

The description of the CCN concentrations in the manuscript differs from the prescribed subsidence in the model setup files (repository https://doi.org/10.5281/zenodo.6514322).

Table 1:
The initial concentration of CCN in the SMT case is listed ad 200 $cm^{-3}$.

Meanwhile, model setup (SMT-RAMSIN) lists the same CCN concentration as in the OLI case.

CCN MAX = 80., ! CCN (#/mg) normally 170

This value also appears in the horizontally averaged concentrations in the model output.

It is of course understandable that authors used similar CCN concentrations in all three cases. However, this has to be clearly stated and also briefly justified.

> We appreciate the reviewer pointing this out, as this was due to an outdated model namelist/data that was uploaded to Zenodo and used to generate some figures. We have corrected the uploaded files (now available at https://doi.org/10.5281/zenodo.6600103), as well as re-generated the affected figures with data from the simulation reported in Table 1. Differences between the simulations were very small and do not affect our conclusions.

**2.2 Subsidence**

The description of the subsidence in the manuscript differs from the prescribed subsidence in the model setup files (repository https://doi.org/10.5281/zenodo.6514322). Lines 231–233: "..Large-scale subsidence is applied throughout the simulation by imposing a horizontal divergence of $2\times10^{-6}s^{-1}$ at every model level, with a boundary condition of $w_{sub}$ = 0 at the surface."

Meanwhile, model setup (ASCOC-RAMSIN, OLI-RAMSIN, and SMT-RAMSIN) do not prescribe this subsidence, but rather large-scale convergence:

CDIVMAX = -6.1224e-6, ! Divergence amplitude (s -1) (negative = convergence)

This large scale convergence also appears in the horizontally averaged vertical velocity in the model output.

This is still and interesting example of an idealized model setup. Nevertheless, the description in the methodology has to be corrected.

The prescribed subsidence is set by the namelist parameter 'DIVLS', not 'CDIVMAX'. The DIVLS values in all namelists correspond to large scale subsidence as correctly stated in the manuscript. CDIVMAX is used for convergence forcing for idealized convection simulations and as such is not used at all in our simulations.

**2.3 Description of Albedo**

The Experiment Setup (2.3) does not describe the surface albedo. While some of the cases take place during the polar day, the surface albedo is an important property in the radiative scheme.

The model setup prescribed albedo value 0.5 (ASCOC-RAMSIN), respectively 0.6 (SMT-RAMSIN). These albedo values should be described in 2.3 and justified.

We have clarified the albedo values used in section 2.3. These values are in-line with albedo measurements taken over the Arctic, which typically range from 0.5-0.7 (e.g. Lindsay and Rothrock, 1994)

**2.4 Comparison Figure not Consistent**

The figure 5 shows an important comparison of all three cases. Unfortunately the time point of CCN removal differ across it figures. The time scales differs as well.

This figure would be much more useful if the time points and time scales were synchronised across all three panels (in a similar way as it is in figure 6).

This figure would furthermore benefit from being larger in size.

The figure has been updated to be aligned vertically, with aerosol dissipation occurring at the same location on each panel and the time scale on either side has been standardized.

**2.5 Turbulent Decay to Non-LES Regime**

In some of the simulations, the dissipation of the clouds leads to decay of turbulence.

This could be clearly seen in the horizontally averaged variable TKEP in the model outputs. By the end of one of the simulations, it drops to values $0.0005$–$0.0009$ $m^2/s^2$, which is almost the same value as in the non-turbulent free atmosphere above the boundary layer (0.0005). This indicates that the subgrid scheme no longer approximated the turbulent cascade and the model no longer resolves large eddies.

While this is totally acceptable result, it would be fair to state in the Discussion and Conclusions that the LES by the end of the simulation does not operate in LES regime, but rather as a fine-scale convection-permitting model.

> In the absence external forcings (which are not applied to our simulations), when the clouds are fully-dissipated there is no mechanisms to generate turbulent motions, so it is natural that TKE settles to the background value.

**2.6 Coupled Boundary Layer**

Line 408:
"...coupled boundary layer (OLI)..."
This statement is in a direct disagreement with the description on line 204: "...surface fluxes are disabled..."

Perhaps it would make sense to correct line 408 to "previously coupled" or "recently decoupled".

> We have changed the language in this line to "previously coupled" as suggested by the reviewer.

**2.7 Strong Conclusion Statement**

Line 422–424:
"We believe that, given the evidence from these three simulations, the microphysical balance state of the cloud may be more important to determining the response to aerosol removal than boundary layer properties"

The conclusions should clarify for which group of boundary layers this applies (decoupled, & polar day, &, large scale convergence, & weak windshear, etc.).

> We have clarified that this conclusion is bounded by the conditions the reviewer has pointed out and that further testing with additional cases, boundary layer conditions, and model forcings is needed to more clearly understand the relative impacts at play.

**2.8–12 Further Minor Comments**

- line 238
  The word "decase" does not seem to fit the context here.

  > Typo fixed to read "decrease"

- Figure 7, panels a and b
  The timeseries "precip" differs between the panels.
  Either there is a numerical mistake, or the captions is incorrect in the sense that "precip" refers to two different variables.

  > This variable refers the to the removal of liquid/ice (for panel (a) and (b), respectively) by accumulation of precipitation at the surface. The caption of the figure has been updated to make this more clear

- Figure 8, panels a and b Same as for Figure 7a,b.

    See response for Figure 7

- The figures 7, 8, and 9 show very important results, but the panels are very small. Please expand them.

    The size of the figures as well as the panels have been increased

- Code and data availability.
  The links should be updated to the newest version of repositories.

    Links have been added

---

## Author Response (AR3)

Point on "2.2 subsidence" - could you please find a way to make sure the same confusion cannot occur to other readers, too?

> We have added a clarification of the namelist variable used when the prescribed subsidence rate is discussed in the manuscript (page 10).

Point on "2.5 Turbulent..." - please consider the statement suggested by the reviewer that for the given cases, the resolution is not quite large-eddy-resolving.

> We have added statement, as suggested by the reviewer, about the lack of LES regime near the end of the simulations to the discussion of the results (page 13).